# Brown adipocytes local response to thyroid hormone is required for adaptive thermogenesis in adult male mice

Yanis Zekri[1]*, Romain Guyot[1], Inés Garteizgogeascoa Suñer[1], Laurence Canaple[1], Amandine Gautier Stein[2], Justine Vily Petit[2], Denise Aubert[1], Sabine Richard[1], Frédéric Flamant[1†], Karine Gauthier[1†]

[1]Institut de Génomique Fonctionnelle de Lyon (IGFL), CNRS UMR 5242, INRAE USC 1370, École Normale Supérieure de Lyon, Lyon, France; [2]U1213 Nutrition, Diabète et Cerveau, Institut National de la Santé et de la Recherche Médicale, Lyon, France

**Abstract** Thyroid hormone (T3) and its nuclear receptors (TR) are important regulators of energy expenditure and adaptive thermogenesis, notably through their action in the brown adipose tissue (BAT). However, T3 acts in many other peripheral and central tissues which are also involved in energy expenditure. The general picture of how T3 regulates BAT thermogenesis is currently not fully established, notably due to the absence of extensive omics analyses and the lack of specific mice model. Here, we first used transcriptome and cistrome analyses to establish the list of T3/TR direct target genes in brown adipocytes. We then developed a novel model of transgenic mice, in which T3 signaling is specifically suppressed in brown adipocytes at adult stage. We addressed the capacity of these mice to mount a thermogenic response when challenged by either a cold exposure or a high-fat diet, and analyzed the associated changes in BAT transcriptome. We conclude that T3 plays a crucial role in the thermogenic response of the BAT, controlling the expression of genes involved in lipid and glucose metabolism and regulating BAT proliferation. The resulting picture provides an unprecedented view on the pathways by which T3 activates energy expenditure through an efficient adaptive thermogenesis in the BAT.

*For correspondence:
yanis.zekri@ens-lyon.fr

†These authors contributed equally to this work

Competing interest: The authors declare that no competing interests exist.

## Editor's evaluation

This valuable manuscript describing the local action of thyroid hormone on brown adipocytes in adaptive thermogenesis in a mouse model is potentially important in advancing our understanding of thyroid hormone action on adipocytes. The experimental approach used in this paper is solid and the claims are convincingly supported by the data. The strength of the data relies on the genetically modified mice developed by the authors and genetic manipulations employed to achieve selective inactivation of TR in adult BAT to arrive at their findings.

## Introduction

Brown adipose tissue (BAT) has the ability to dissipate energy through thermogenesis in response to cold and high-fat diet (HFD), to prevent hypothermia and limit body weight gain, respectively. In response to these stressors, sympathetic nerves stimulate brown adipocytes' adrenergic receptors to trigger the cAMP-protein kinase A (PKA) (*Tabuchi and Sul, 2021*), favoring local lipolysis and glycolysis to fuel an increase in energy metabolism (*Hankir and Klingenspor, 2018*). This is used by the BAT-specific protein UCP1 to favor thermogenesis at the expense of ATP production (*Fenzl and Kiefer, 2014*). In the long run, this causes an increase in lipogenesis to refill local lipid stocks, mitochondrial

biogenesis, and a proliferation of brown adipocytes (*Fonseca et al., 2014*; *Uldry et al., 2006*; *Fukano et al., 2016*). These actions are coordinated by transcription factors, notably involving the coactivator PGC1α (*Cao et al., 2004*). Importantly, cold exposure induces an indispensable increase expression and activity of the type 2 deiodinase activity (DIO2) resulting in an intracellular conversion of the inactive form of thyroid hormone, thyroxine or T4, into its active form, 3,3′,5-triiodo-L-thyronine or T3 (*Fonseca et al., 2014*; *Silva and Larsen, 1985*; *Bianco and Silva, 1987*).

T3 exerts a large influence on energy expenditure, as hypothyroidism is associated with cold sensitivity and weight gain, while hyperthyroid patients display the opposite phenotype (*Bianco and Silva, 1987*; *Wolf et al., 1996*; *De Leo et al., 2016*; *Maushart et al., 2019*). T3 binds to the broadly expressed thyroid hormone nuclear receptors, TRα1 and TRβ1/2 (collectively called TR) encoded by the *Thra* and *Thrb* genes (*Minakhina et al., 2020*). Majority of TR bind to specific response elements resembling the archetypical DR4 consensus sequence (5′AGGTCAnnnnRGGnCA3′) to repress expression of neighboring genes. Upon T3 binding, TR results in a rapid activation of transcription (*Flamant, 2016*).

T3 concentration drastic increase in BAT during cold exposure (*Bianco and Silva, 1988*) is crucial, as DIO2*KO* mice have impaired BAT thermogenesis (*Schneider et al., 2001*). This was notably explained by the ability of T3 to regulate essential thermogenic processes in the BAT-like *Ucp1* expression (*Bianco et al., 1988*) and fatty acids oxidation (*Fonseca et al., 2014*). However, the involvement of T3 in energy metabolism is not restricted to brown adipocytes, as it directly increases hepatic lipid and glucose metabolism (*Ritter et al., 2020*), and favors exercise-associated thermogenesis in the muscle (*Nicolaisen et al., 2020*). It also triggers white adipose tissue (WAT) browning, a process in which heat-producing 'beige' adipocytes expressing *Ucp1* emerge in the WAT (*Martínez-Sánchez et al., 2017*; *Medina-Gomez et al., 2008*). Finally, T3 stimulates indirectly BAT thermogenesis by acting in the hypothalamic ventro-medial nuclei where it triggers the sympathetic outflow to BAT (*López et al., 2010*).

Thus, T3 exerts a broad influence on different tissues to regulate energy expenditure, making the relative importance of BAT thermogenesis in T3-dependent energy expenditure unclear. Moreover, T3 action directly in the BAT has not been exhaustively described as (1) no transcriptomic and cistromic analyses have been led to obtain T3/TR target genes in this tissue and (2) T3 also indirectly controls the BAT through the brain, making the role of local T3 difficult to pinpoint. Here, we aimed at distinguishing these two levels and focus on the T3 cell-autonomous (i.e., local) function in brown adipocytes, and address its importance in adaptive thermogenesis. To address this aim, we combined genomic analysis and the phenotyping of mice with selective blockade of T3 signaling in brown adipocytes. The resulting picture provides an unprecedented view on the pathways by which T3 activates energy expenditure through an efficient adaptive thermogenesis in the BAT.

## Results

### BATKO mice present a BAT-specific deletion of T3 signaling

The blockade of T3 signaling specifically in brown adipocytes in adults has been achieved using the *Ucp1*^CreERT2^ (*Figure 1—figure supplement 1*; *Rosenwald et al., 2013*). It was combined with two available floxed alleles to ascertain a full blockade of the T3 response in brown adipocytes. In the absence of tools to eliminate *Thra*, we used *Thra*^AMI^, a recombinant 'floxed' allele of the *Thra* gene, in which Cre-mediated recombination allows the expression of TRα1^L400R^, a dominant-negative version of the TRα1 receptor (*Quignodon et al., 2007*). TRα1^L400R^ prevents the recruitment of coactivators and is thus condemned to constitutively repress T3 target genes expression. Thus, we expect the effects of this knock-in to be stronger than a knock-out. Then, we used *Thrb*^lox^, a recombinant allele of *Thrb* gene, in which two loxP sequences flank the exon encoding the DNA-binding domain of TRβ1/TRβ2 (*Winter et al., 2009*). This approach eliminates T3 responsiveness. *Ucp1*^CreERT2^x*Thra*^AMI/+^*Thrb*lox^lox/lox^ mice (*Figure 1A*) are called BATKO mice and compared with *Thra*^AMI/+^*Thrb*lox^lox/lox^ CTRL littermates.

We verified that the *Thra*^AMI^ allele was expressed in the BAT of BATKO mice (*Figure 1B*) and that *Thrb* expression was drastically reduced (93%) in the BAT of BATKO mice, but not in other tissues (*Figure 1C*). As expected, the T3-induced regulation of *Hr* expression, a classical T3 target gene in many tissues (*Zekri et al., 2022*), was selectively and almost completely lost in the BAT of BATKO mice (*Figure 1D*). The observed residual responses most likely reflect the presence in BAT of other cell

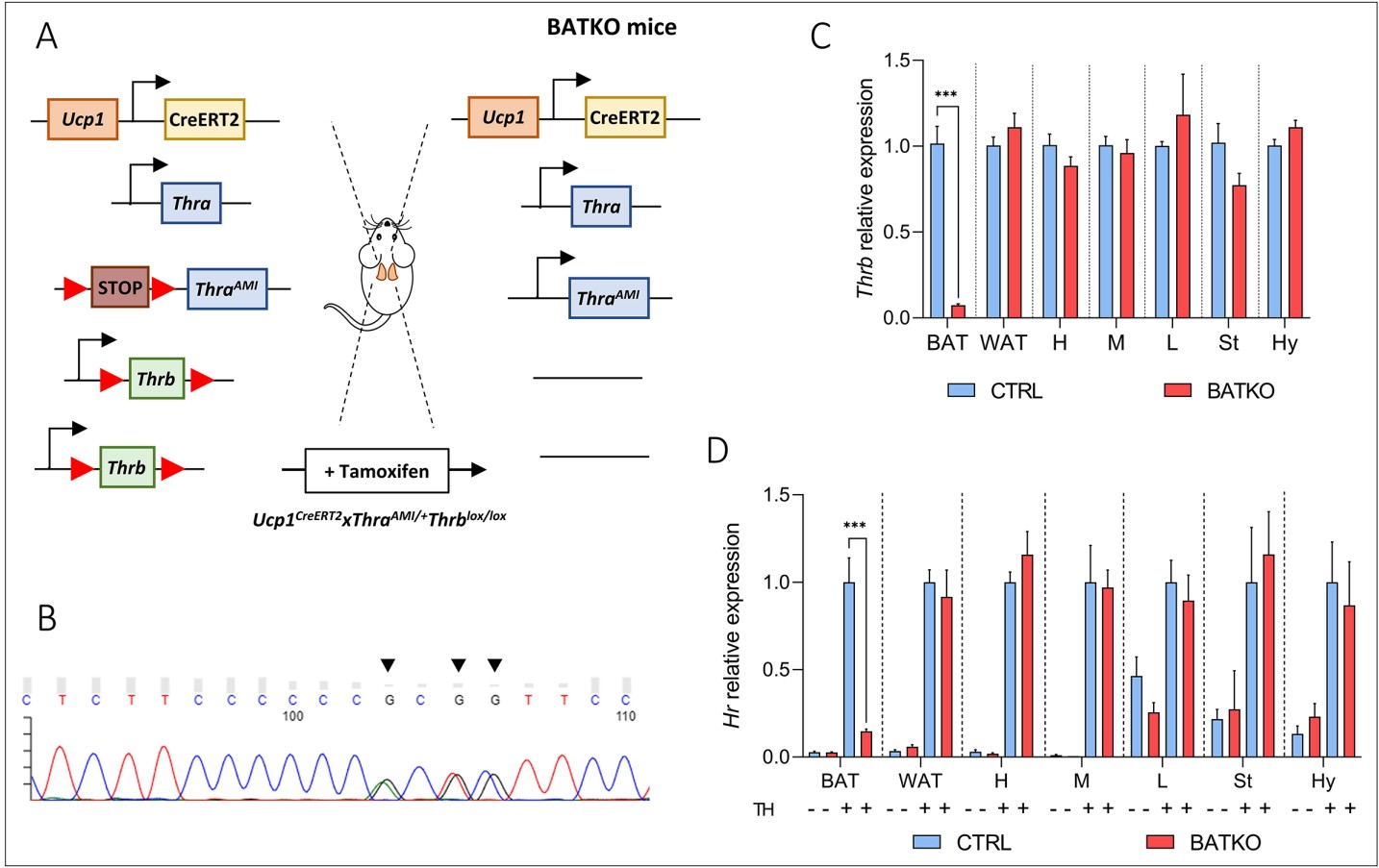

**Figure 1.** BATKO mice present a brown adipose tissue (BAT)-specific blockade of T3 signaling. (**A**) Schematic representation of the BATKO mice. BATKO mice carry the *Ucp1CreERT2* transgene, allowing the brown-adipocyte-specific expression of the tamoxifen-sensitive CreERT2 recombinase. BATKO mice are also heterozygous for the *ThraAMI* allele, which encodes the TRα1L400R dominant-negative receptor after Cre-mediated deletion of a STOP cassette flanked by loxP sequences. BATKO mice are homozygous for the *Thrblox* allele in which exon 3 is flanked by two tandem-arranged loxP sequences. After tamoxifen injection, Cre-mediated recombination selectively excise the loxP-flanked sequences in brown adipocytes, resulting in the expression of TRα1L400R and elimination of TRβ. Control mice (CTRL, not represented here) had the same genotype except for the absence of the *Ucp1CreERT2* transgene and were also tamoxifen treated. (**B**) Sanger sequencing chromatogram of a fragment of *Thra* cDNA prepared from BAT RNA of BATKO mice. Arrows indicate the positions of the TRα1L400R mutations. (**C**) Relative mRNA expression of *Thrb* in different peripheral and central tissues of CTRL and BATKO mice (H: heart, M: muscle, L: liver, St: striatum, Hy: hypothalamus) (n = 5–6/group). (**D**) Evaluation of T3 response in propylthiouracil (PTU)-fed CTRL and BATKO mice through the induction of *Hr*, a well characterized TR target gene, after 24 hr of TH (+ or −; n = 5–7/group). Statistical significance is shown for the comparison of CTRL and BATKO mice treated with TH. Error bars represent the standard deviation (SD). \*\*\*p < 0.001 for the indicated comparisons.

The online version of this article includes the following figure supplement(s) for figure 1:

**Figure supplement 1.** CRE activity in peripheral and central tissues.

**Figure supplement 2.** Serum concentration of TH in CTRL and BATKO mice.

types, like endothelial cells and immune cells (*Biagi et al., 2021*). Importantly, there was no significant difference between BATKO and CTRL mice in free T3 or free T4 serum levels, whatever the temperature and diet conditions used in this report (*Figure 1—figure supplement 2*). Altogether, BATKO mice present an altered T3 signaling specifically in brown adipocytes, allowing us to decipher the cell-autonomous functions of T3 in this cell type.

## Establishment of a catalog of TR direct target genes in BAT

We first aimed at establishing a complete list of TR direct target genes in brown adipocytes, defined as genes: (1) with a TR-binding site (TRBS) within 30 kb of the transcription start site (TSS) (*Chatonnet et al., 2013*), (2) which mRNA levels are rapidly (within 24 hr) increased in BAT in response to T3 and T4 (collectively TH), and (3) which induction is lost in BATKO mice, that is, controlled locally by TR in

brown adipocytes (and thus, not secondary to the sympathetic stimulation of the tissue caused by the TH treatment).

*Ucp1^{CreERT2}xThra^{GS/+}* mice express specifically in brown adipocytes a GS-tagged version of TRα1 (*Hirose et al., 2019*; *Richard et al., 2020*) used for chromatin immunoprecipitation sequencing (ChIPseq). We identified 4210 TRBS, in the vicinity of 2311 genes, essentially located within 10 kb of the TSS (*Figure 2A*) and mostly in intronic sequences (*Figure 2B*). TRBS was preferentially present on motifs related to the so-called DR4 consensus sequence (AGGTCAnnnnRGGnCA), described as preferential for TR fixation (*Flamant, 2016*; *Figure 2C*).

Time-course analysis of BAT transcriptome in hypothyroid (PTU-fed) wild-type mice treated with TH for 3, 6, 12, or 24 hr was conducted by RNAseq. It revealed that a large number of genes were regulated (1946 upregulated, 1744 downregulated) in a time-dependent manner (*Figure 2D*). We observed that 49% of genes induced by T3 after 3 hr possess a TRBS within 30 kb of their TSS. This ratio fell below 10% for T3-repressed genes and for genes which expression was insensitive to T3 (*Figure 2E*). This suggests that downregulation of gene expression is not directly exerted by T3-bound TR, but is an indirect consequence of the TH treatment, either cell autonomous or resulting from sympathetic stimulation. Given these considerations, the rest of the study was restricted to positively regulated genes.

We then crossed the RNAseq and the ChIPseq datasets to obtain a curated list of 639 putative TR direct target genes (*Figure 2F*, *Figure 2—source data 1*), whose expression was most likely under the direct control of T3/TR signaling. As expected, this set of genes included two emblematic TR target genes: *Ucp1* (*Martinez de Mena et al., 2010*) and *Hr* (*Engelhard and Christiano, 2004*; *Figure 2G*). Heatmap representation showed that TR direct target genes previously found to be upregulated after 24 hr of TH treatment in wild-type mice were not responsive in BATKO mice (*Figure 2H*). Transcriptome results were confirmed by RT-qPCR for some of the identified TR direct targets (*Figure 2I*). We hypothesized that the residual response to T3 of BAT in BATKO mice might result either from the stimulation of brown adipocytes by the sympathetic system (*López et al., 2010*) or from the T3 response of BAT cells which do not express *Ucp1*, like endothelial or immune cells. To distinguish between these two possibilities, we chemically destroyed BAT noradrenergic terminals of wild-type PTU-fed mice (*Figure 2—figure supplement 1*, *Figure 2—source data 2*), treated or not afterwards with TH. T3-induced expression of target genes was mostly similar between sham and denervated mice (*Figure 2J*), indicating that, for the genes that we tested, the sympathetic stimulation was not involved.

According to gene ontology analysis, TR direct target genes were directly involved in the 'regulation of cold-induced thermogenesis' (*Figure 2K*), including *Ucp1* and *Ppargc1a*, two fundamental actors of this process. Interestingly, *Ppargc1a* encodes for PGC1α, a co-activator of TR (*Yuan et al., 2013*). Using a published list of PGC1α-binding sites (GSE110053) (*Chang et al., 2018*), we found that around 33% of TR direct target genes showed a co-localization of TR and PGC1α-binding sites (*Figure 2—figure supplement 2*), including genes involved in lipid metabolism as well as *Ucp1* and *Ucp3* (*Figure 2—source data 3*). In addition, many of TR direct target genes we identified were involved in mitochondrial transport, respiratory chain, catabolism, and biogenesis of lipids, as well as genes involved in the glycolysis and the citric acid cycle. Finally, we also found genes involved in proliferation, a process of BAT mid-term adaptation to physiological stressors like cold (*Fukano et al., 2016*). In summary, TR direct target genes belong to several biological processes, many of them being directly connected to BAT thermogenesis.

## Altered response of BATKO mice to temperatures below thermoneutrality

Based on the roles of TR target genes identified above, we predicted that BATKO mice would display alterations in BAT thermogenesis. At 23°C, neither body weight, nor body composition, nor metabolic rate were altered in BATKO mice, but food consumption was increased (*Figure 3—figure supplement 1*). As these small variations occurred at 23°C, which is a moderate cold exposure for mice (*Reitman, 2018*), we submitted them to a more drastic cold challenge.

BATKO mice maintained their body temperature normally at 4°C during 72 hr but they tended again to consume more food than CTRL mice (*Figure 3A*). We thus combined cold exposure with fasting, causing a severe hypothermia in BATKO mice which led us to end the experiment (*Figure 3B*). A similar phenotype was obtained when TRβ was the only mutated TR, suggesting that a significant

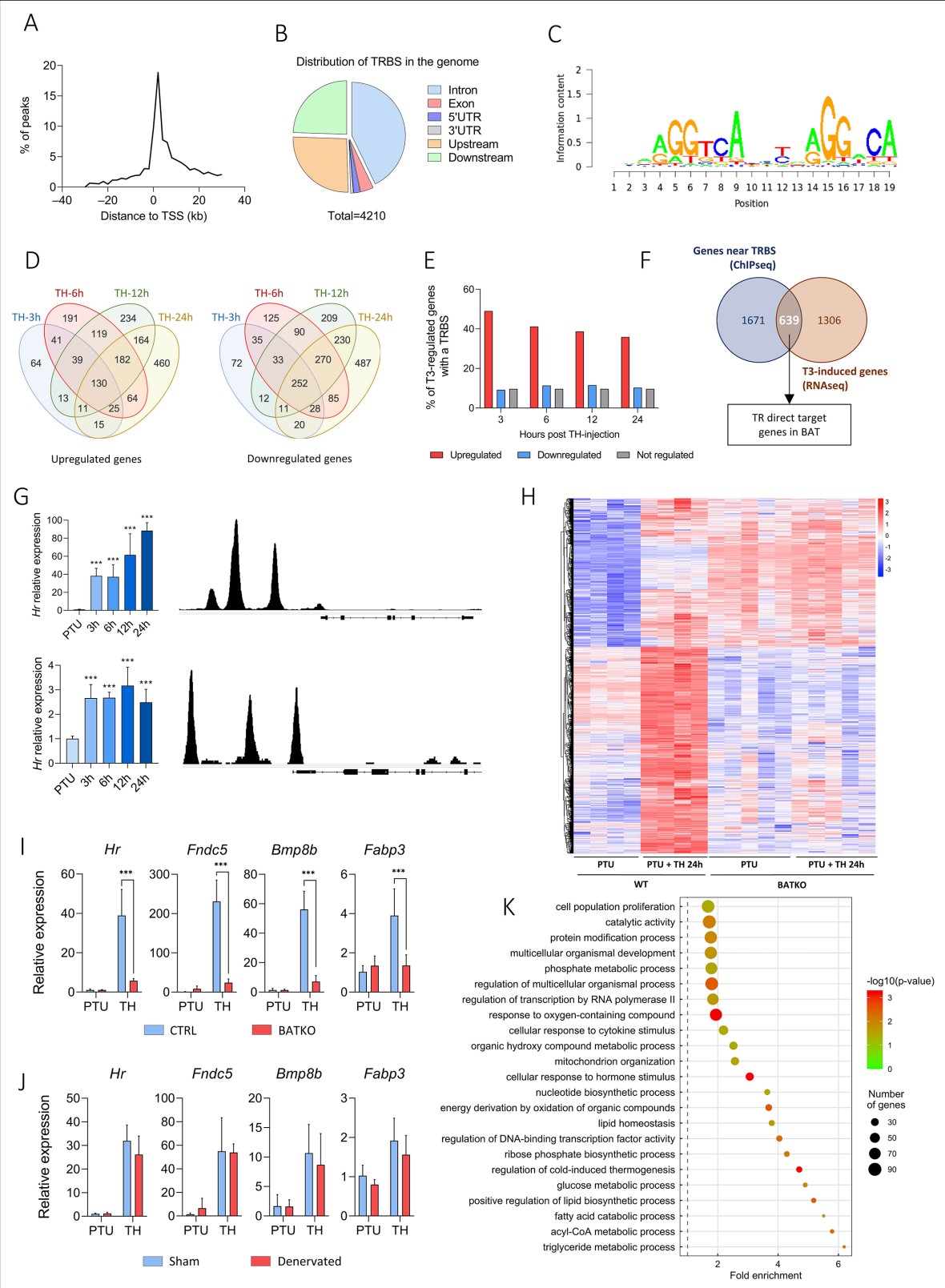

**Figure 2.** Identification of T3/TR target genes in brown adipocytes. (**A**) Consensus sequence found in brown adipose tissue (BAT) TRBS (thyroid hormone receptor-binding site), as identified by de novo motif search. (**B**) Frequency of TRBS distribution around transcription start sites (TSSs). (**C**) Pie chart of the 4210 TRBS distribution in the genome (UTR: untranslated region). (**D**) Venn diagrams of upregulated (left panel) and downregulated (right panel) genes after TH intraperitoneal injection in wild-type propylthiouracil (PTU)-treated mice for different periods. (**E**) Percentage of genes which

*Figure 2 continued on next page*

*Figure 2 continued*

possess a TRBS within 30 kb of their TSS among genes whose expression in the BAT is regulated or not by T3 (upregulated in red, downregulated in blue, not regulated in gray). (**F**) Venn diagram of genes whose expression is induced by T3 in at least one of the time points (in brown, RNAseq data) and genes with a TRBS within 30 kb of their TSS (in blue, chromatin immunoprecipitation sequencing [ChIPseq] data), that is, TR direct targer genes. (**G**) Left: Time-course analysis of *Hr* (top) and *Ucp1* (bottom) expression in BAT after 24 hr of TH treatment of wild-type hypothyroid mice (RT-qPCR). Statistical significance is shown for the different time points versus untreated PTU-fed mice (*n* = 4–6/group). Right: Extract of the TRBS in the *Mus musculus* genome browser around *Hr* (top) and *Ucp1* (bottom). (**H**) Heatmap representing in both CTRL and BATKO mice the expression of TR direct target genes upregulated after 24 hr of TH injection in CTRL hypothyroid mice. Colors represent the *z*-scores, see scale besides the heatmap (*n* = 4–5/group). Relative expression of several TR direct target genes 24 hr after TH treatment in (**I**) CTRL/BATKO and (**J**) Sham/denervated PTU-fed C57BL6/J mice. Statistical significance is shown for the comparisons between CTRL-TH and BATKO-TH, or SHAM-TH and BATKO-TH mice (*n* = 5–7/group). (**K**) Gene ontology dot plot of the 639 TR direct target genes. Only the 'biological processes' terms with a fold-enrichment >1.5 were kept. Some of the terms were shortened to increase readability without affecting the meaning. Error bars represent the standard deviation (SD). ***p < 0.001 for the indicated comparisons.

The online version of this article includes the following source data and figure supplement(s) for figure 2:

**Source data 1.** Raw data and compilation of T3/TR target genes in brown adipocytes.

**Source data 2.** Raw western blots for brown adipose tissue (BAT) denervation confirmation.

**Source data 3.** List of T3/TR target genes that also include PGC1α-binding sites.

**Figure supplement 1.** Confirmation of brown adipose tissue (BAT) denervation by quantification of tyrosine hydroxylase.

**Figure supplement 2.** Comparison of TR direct target genes with the genes presenting a PGC1α-binding site.

part of the effects mediated by T3 in BAT requires TRβ (*Figure 3—figure supplement 2*). Globally, alteration of T3 signaling in BAT requires a higher energy intake to maintain body temperature during a cold stress. This suggests that compensatory thermogenic processes are triggered to maintain body temperature in BATKO mice fed ad libitum. In that context, we notably observed that the browning of the WAT was exacerbated in BATKO mice (*Figure 3—figure supplement 3*).

HFD represents another challenge for the thermogenic capacity. BATKO mice gained less weight than CTRL mice during HFD at 23°C, despite similar food intake (*Figure 3C*). Again, this phenotype was reproduced when TRβ only was mutated, reinforcing the importance of TRβ in T3-mediated regulation of BAT thermogenic processes (*Figure 3—figure supplement 2*). Collectively, the resistance to diet-induced obesity suggests that mice with an altered T3 signaling in BAT have a higher energy expenditure. Higher energy expenditure was not observed in BATKO mice by 48 hr of indirect calorimetry (*Figure 3—figure supplement 1*) but even a minor difference could explain the subtle difference between BATKO and CTRL mice. HFD feeding was then conducted at thermoneutrality, eliminating the need to activate alternate mechanisms to defend body temperature. In this condition, the opposite result was observed, BATKO mice being more sensitive to diet-induced obesity than CTRL mice with similar food intake (*Figure 3D*). Collectively, these results point out that BATKO mice suffer from a reduced efficiency of BAT adaptive thermogenesis both in condition of cold exposure and excess of calories. When exposed to HFD at 23°C, the activation of alternative thermogenic processes combined with the defect in adipocytes thermogenesis results in a paradoxical resistance to obesity of BATKO mice.

## BAT TR signaling controls the expression of a subset of genes induced during cold exposure

To better understand the molecular response controlled by TR signaling in the BAT during cold response, we compared the BAT transcriptome of BATKO and CTRL mice after 24 hr at 4°C, in the presence of food. Among 2865 cold-induced genes (*Figure 4—source data 1*), 491 (17%) displayed a different response in BATKO mice (*Figure 4A*). For most of them, the cold induction was partially or completely lost in BATKO mice. Noteworthy, *Ppargc1a* was the only classical thermogenic marker that was part of this subset (*Figure 4—source data 1*). The highly stringent statistical interaction model that we used (*Love et al., 2014*) failed to reveal an influence of T3 on the cold response of the *Ucp1* and *Dio2* genes, two classical thermogenic markers. Thus, we used RT-qPCR to measure *Ucp1* and *Dio2* mRNA levels on a larger number of samples and found that the cold induction of these genes was indeed altered in BATKO mice (*Figure 4B*). Thus, the high stringency of the statistical model used avoids false positives, allowing to trustfully highlights genes of interest, but some of them can be missed due to a lack of statistical sensitivity.

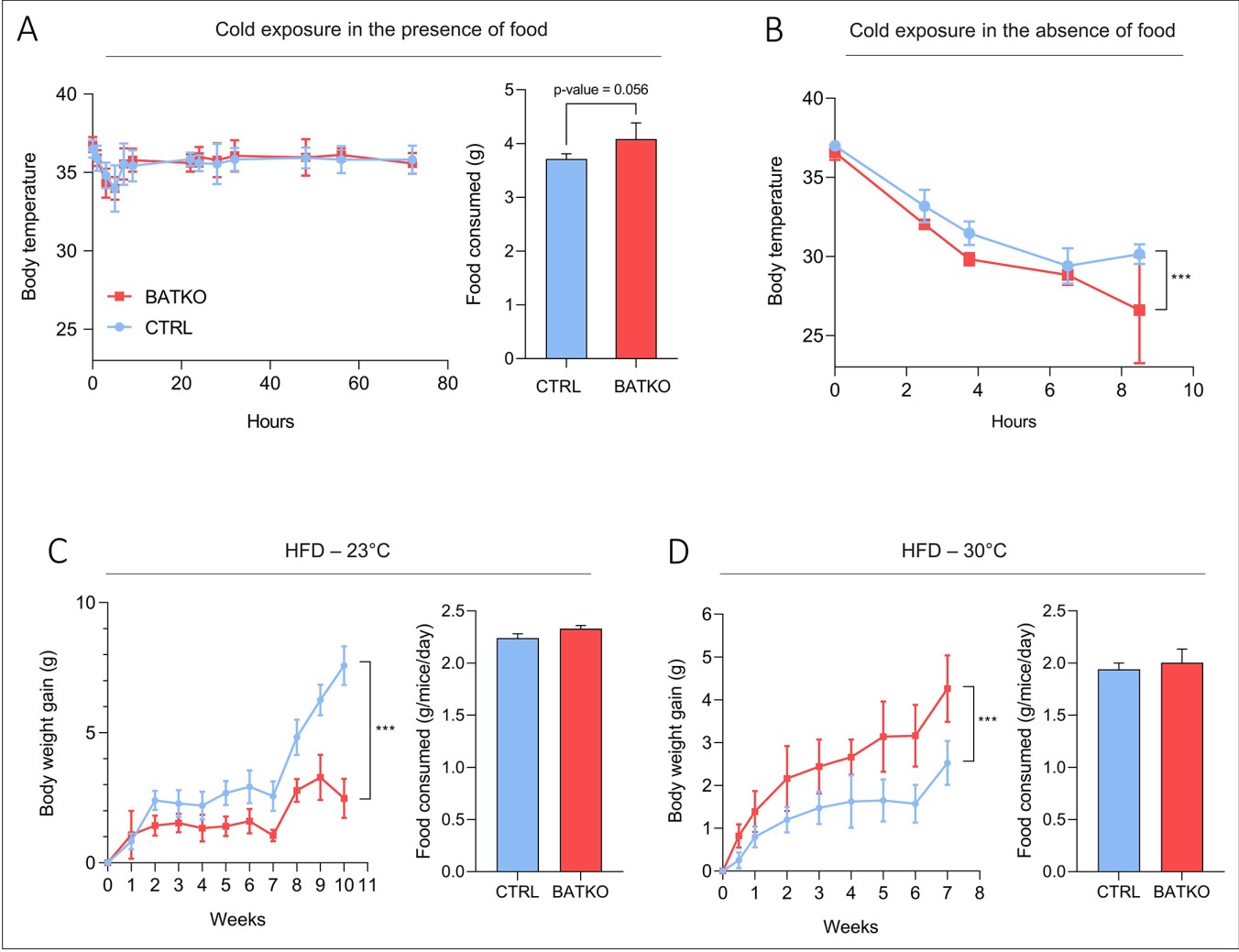

**Figure 3.** BATKO mice are cold sensitive. (**A**) Core body temperature (left) and food consumption (right, over 48 hr) of CTRL and BATKO mice exposed to 4°C in the presence of food (n = 5–7/group). BATKO mice tended to eat more than CTRL mice. (**B**) Core body temperature in CTRL and BATKO mice exposed to 4°C in the absence of food. After 8 hr of cold exposure, BATKO mice reached severe hypothermia and the experiment was stopped (n = 4–5/group). Body weight gain and food consumption of CTRL and BATKO mice at room temperature (**C**, n = 4–5/group) or 30°C (**D**, n = 5–8/group). Error bars represent the standard error of the mean. ***p < 0.001.

The online version of this article includes the following figure supplement(s) for figure 3:

**Figure supplement 1.** Body composition and indirect calorimetry in CTRL and BATKO mice.

**Figure supplement 2.** BATβKO mice are also cold sensitive.

**Figure supplement 3.** White adipose tissue (WAT) browning is exacerbated in BATKO mice.

Using gene ontology, we found that the genes which cold response was significantly altered in BATKO mice were often directly connected to thermogenesis (***Figure 4C***). This gene set notably included genes involved in mitochondrial activity and respiratory chain. Other genes were involved in glycolysis, Krebs cycle, lactate metabolism, and glucose transport which have all been shown to be crucial to fuel BAT thermogenesis (***Jeong et al., 2018***). Genes involved in both lipolysis/fatty acid oxidation and lipogenesis, two processes required for an appropriate lipid use during thermogenesis, were also altered. Finally, a gene set reflecting cell proliferation was activated (***Figure 4—source data 2***). Restriction of this list to the TR direct target genes (***Table 1***) also highlighted secreted peptides (*Bmp8b* and *Fgf21*) and enzymes participating in heat-producing creatine futile cycle (*Alpl*). *Ppargc1a* still belonged to this list of restricted genes of interest, reinforcing its importance in TR-mediated regulation, as abovementioned. Collectively, the BAT transcriptome of BATKO mice evidenced several altered pathways crucial for BAT thermogenesis that could explain their phenotype.

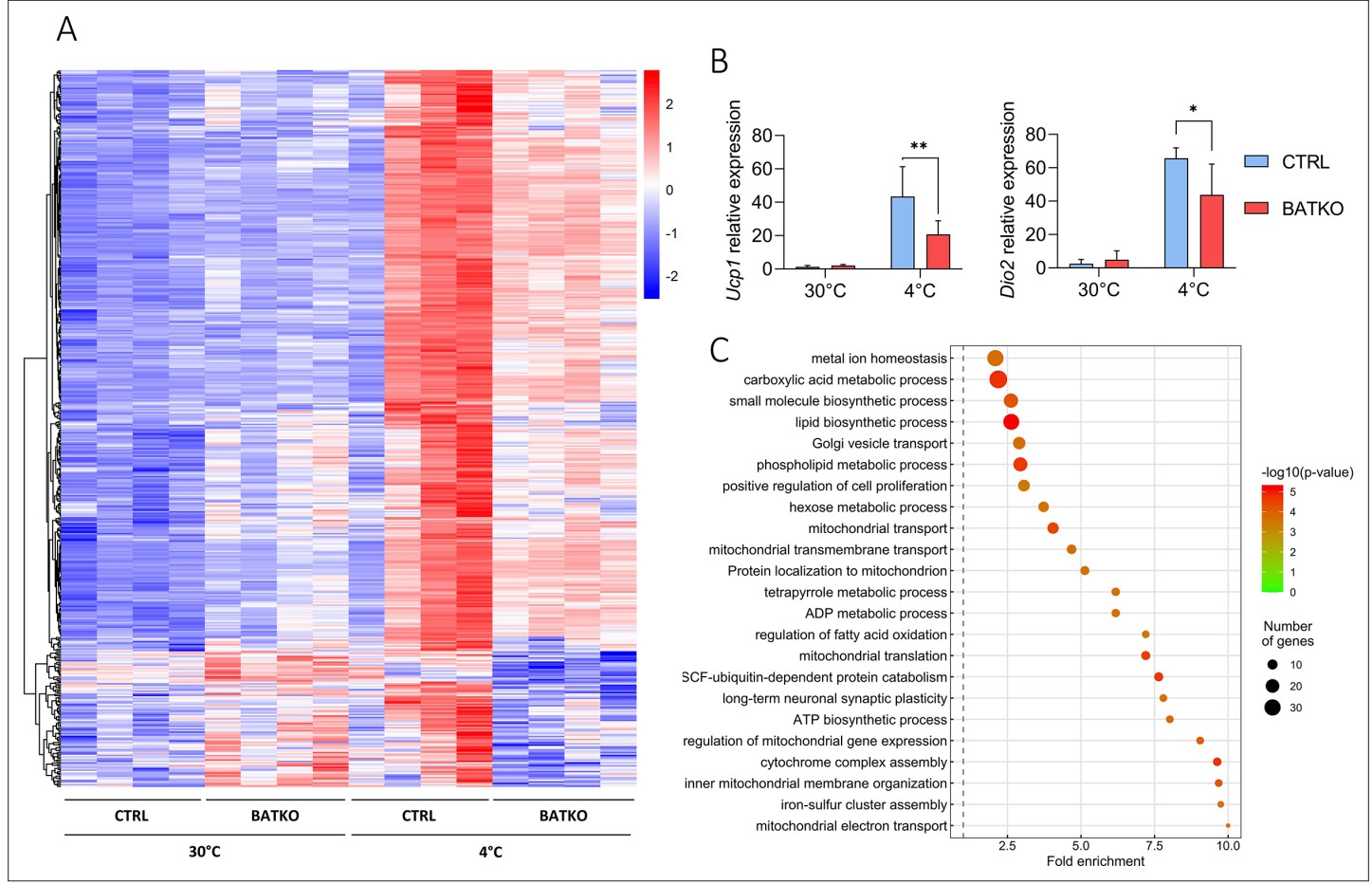

**Figure 4.** A subset of T3-regulated genes is activated during cold exposure and is necessary for an efficient brown adipose tissue (BAT) thermogenic response. (**A**) Heatmap representation of cold-responsive genes altered in BATKO mice, after 24 hr at 4°C. Colors represent the z-scores, see scale in heatmap (n = 3–5/group). (**B**) Relative mRNA expression of *Ucp1* (left panel) and *Dio2* (right panel) in CTRL and BATKO mice at 30°C or after 24 hr at 4°C. Statistical significance is shown for the comparison between CTRL and BATKO mice at 4°C (n = 4–6/group). (**C**) Gene ontology dot plot representation of biological processes enriched in the 491 genes inefficiently induced in BATKO mice at cold. Some of the terms were shortened to increase readability without affecting the meaning. Error bars represent the standard deviation (SD). *p < 0.05, **p < 0.01 for the indicated comparisons.

The online version of this article includes the following source data for figure 4:

**Source data 1.** Transcriptome analysis of cold response in the brown adipose tissue (BAT) of both CTRL and BATKO mice.

**Source data 2.** List of T3/TR target genes which expression is affected in the brown adipose tissue (BAT) of BATKO mice during cold exposure.

**Table 1.** List of T3/TR target genes dysregulated during cold exposure in BATKO mice.

| Biological function | TR-dependent cold-induced genes |
|---|---|
| Lipid metabolism | *Fabp3, Pla2g12a, Acsl5, Aspg, Mcee* |
| Glucose metabolism | *Slc2a4, Ogdh, Idh3a* |
| Cell cycle progression | *Cux1, Ccnd1* |
| Mitochondrial activity | *Ppargc1α, Coq10a, Uqcc2, Cyb561* |
| Futile cycle | *Alpl* |
| Secreted peptides | *Bmp8b, Fgf1* |

## T3 in BAT controls BAT proliferation

RNAseq analysis revealed that TH might participate in BAT proliferation, both during hyperthyroidism and cold exposure. After 5 days of TH treatment, wild-type PTU-fed mice displayed an overexpression of 72% of a group of genes previously described to be involved in cell cycle (*Whitfield et al., 2002*; *Figure 5—figure supplement 1*). This translated into an effective increase in cell proliferation when TH treatment was associated with an injection of EdU 24 hr before sampling. This effect was insensitive to BAT denervation

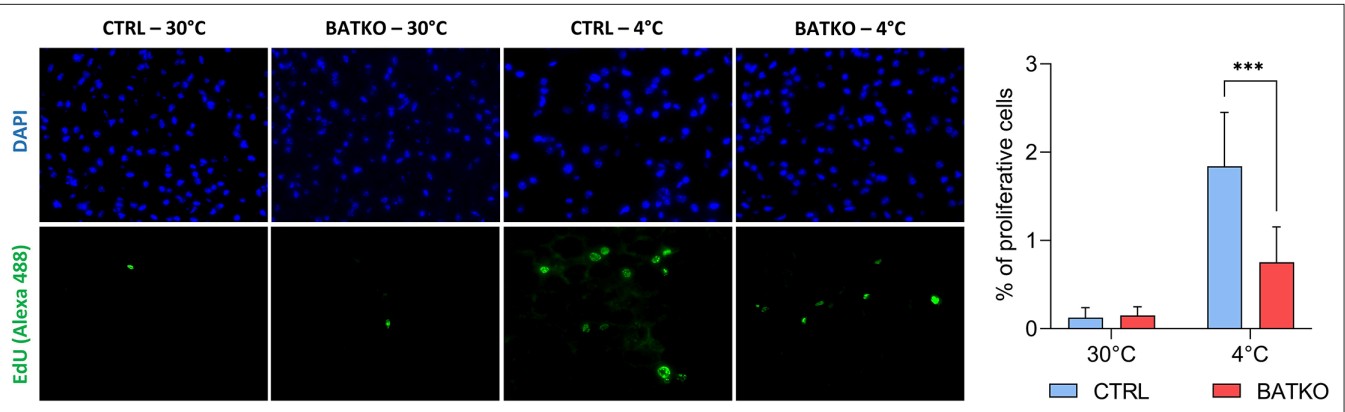

**Figure 5.** Local control of brown adipocytes proliferation by T3. Representative images (left) and quantification (right) of EdU-positive proliferative cells in brown adipose tissue (BAT) from both CTRL and BATKO mice exposed 72 hr to 4°C and injected with EdU after 24 and 48 hr of cold. Nuclei are stained in blue with DAPI. Percentage of proliferative cells is the ratio of proliferative cells on the number of nuclei (n = 4–7/group). Statistical significance is shown for the comparison CTRL 4°C versus BATKO 4°C. Error bars represent the standard deviation (SD). ***p < 0.001r for the indicated comparisons.

The online version of this article includes the following figure supplement(s) for figure 5:

**Figure supplement 1.** TH trigger brown adipose tissue (BAT) proliferation.

(*Figure 5—figure supplement 1*), suggesting that it most likely results from a cell-autonomous response to T3.

As predicted by the transcriptomic data after 24 hr of cold exposure, we also observed that proliferation is triggered in the BAT when CTRL mice were exposed 72 hr to cold, but 2.5-fold less proliferative cells were observed in BATKO mice, in the same conditions (*Figure 5*).

As most of the cell-cycle genes induced by T3 do not have a TRBS within 30 kb (*Figure 5—figure supplement 1*), the link between the TR direct target genes and proliferation in BAT yet remains uncertain. *Ccnd1*, encoding cyclin D1, is one of the TR direct target gene (*Figure 2—source data 1*) and also belongs to the genes that are under the control of TH signaling during cold exposure (*Figure 4—source data 2*). It is thus an interesting candidate to play a significant part in this process.

## Discussion

T3 has been for a long time under the lights of metabolism research for its ability to regulate energy expenditure in many tissues, including the BAT. However, neither the exhaustive list of TR direct target genes in BAT, nor the contribution of BAT in T3-mediated regulation of energy expenditure have been elucidated so far. Here, we present an unprecedented in-depth analysis of T3 direct influence on gene expression in brown adipocytes and developed a transgenic mice model with a brown-adipocyte-specific suppression of T3 signaling. To our knowledge, this model is currently the only one allowing a suppression of T3 signaling specifically in brown adipocytes at adult stages. This is a significant step forward as it allows (1) to dissect the contribution of BAT in the T3-mediated increased in energy expenditure, (2) to distinguish the hypothalamic (*López et al., 2010*) versus local control of BAT thermogenesis, and (3) to bypass any developmental alteration mediated by the early lack of T3 signaling (*Hall et al., 2010*). Collectively, this new model showed that T3 signaling was pivotal for BAT adaptive thermogenesis, regulating the management of thermogenic fuels as well as the plasticity of the tissue.

Combined with transcriptome analyses, the genome-wide study of TRα1 chromatin binding allowed us to identify 639 genes whose transcription is most likely controlled by liganded TR in brown adipocytes. This set of genes only partially overlaps with the ones described in other cell types, probably as a result of differential chromatin occupancy (*Chatonnet et al., 2013*; *Hirose et al., 2019*; *Richard et al., 2020*) and the presence of cell-specific transcription cofactors. One of these cell-specific cofactors is PGC1α (encoded by *Ppargc1a*) which we found here to be a TR direct target gene. As PGC1α is a transcriptional coactivator of TR (*Yuan et al., 2013*), it confirms previous hypothesis that TR and PGC1α are involved in an auto-regulatory feed-forward loop (*Wulf et al., 2008*). In line with this, we

found that 33% of the TRBS in brown adipocytes are also occupied by PGC1α, notably for genes involved in lipid metabolism or directly involved in thermogenesis, like *Ucp1* and *Ucp3* (*Cannon and Nedergaard, 2004*; *Silvestri et al., 2020*). PGC1α is thus likely to play a pivotal role in the T3-dependent regulation of energy metabolism in brown adipocytes. As PGC1α is also a coactivator of several other nuclear receptors, notably PPARγ and ERRα, its overexpression might generate a cross-talk with other signaling pathways.

One of the main challenges of our study was to make a distinction between a local influence of T3 and an indirect consequence, resulting from the hypothalamic response to T3 (*López et al., 2010*). The BAT response to T3 is largely lost in BATKO mice and conserved in denervated mice, suggesting that the changes in gene expression observed in BAT mainly reflect the cell-autonomous response initiated by TR in brown adipocytes and not the effect of T3 mediated by the hypothalamus (*López et al., 2010*). Residual responses to T3 in BATKO may reflect the sensitivity of other cells types present in the BAT which do not express the *Ucp1*$^{CreERT2}$ transgene or an incomplete effect mediated by tamoxifen. Otherwise, they could be secondary to a modification of circulating peptides caused by the T3 treatment on other organs. For instance, *Fgf21* expression is induced by T3 in the liver (*Adams et al., 2010*) and the resulting secreted peptide has been shown to promote BAT thermogenesis (*Sáenz de Urturi et al., 2022*). We found indication for a reciprocal influence as several TR direct target genes in brown adipocytes encode secreted factors and adipokines (*Apln*, *Fdcn5*, and *Bmp8b*), that are able to act in other organs. This suggests that T3 can influence a complex network of cross-talks in the organism.

T3 signaling in BAT is triggered in CTRL mice during cold exposure as it was previously shown by the activation of type 2 deiodinase activity in similar conditions (*de Jesus et al., 2001*). This activation of T3 signaling represents a significant fraction of the cold response, since 17% of the gene regulations induced in the BAT by cold exposure in CTRL mice are lost in BATKO mice. We showed that T3 signaling in the BAT controls the expression of genes crucial for both lactate metabolism (*Ldha*) or glycolysis (*Hk1*, *Aldoa*, *Pkm*, and *Pfkl*), two pathways essential for BAT thermogenesis (*Jeong et al., 2018*). T3 also controls genes involved in lipid metabolism, which is also congruent with previous observations revealing that T3 signaling is required for efficient lipogenesis (*Christoffolete et al., 2004*) and lipolysis (*Oppenheimer et al., 1991*). Here, we bring a broader picture by identifying all the genes which expression is regulated by T3 for these processes.

Interestingly, we showed that *Ppargc1a* was among impacted genes. As it coactivates other nuclear receptors, we can imagine that part of the genes dysregulated in BATKO mice can be attributed to other signaling pathways downstream of PGC1α. On the other hand, we observed that other classical thermogenic markers were not the most obviously affected in BATKO mice. Indeed, *Ucp1* and *Dio2* could not be detected as differentially expressed by our stringent whole-transcriptome approach but only by targeted expression assessment. Among TR-dependent cold-induced genes, we rather detected *Alpl*, a phosphatase required in brown adipocytes for UCP1-independent thermogenesis derived from futile creatine cycle (*Sun et al., 2021*). Thus, T3 signaling in BAT coordinates the expression of genes involved in both UCP1-dependent and UCP1-independent thermogenesis. Finally, we showed that T3 signaling in brown adipocytes controls cell proliferation both during cold exposure and upon hyperthyroidism. This concurs with a recent study showing that T3 promotes BAT proliferation in adult mice in a cell-autonomous manner (*Liu et al., 2022*). Also, the link between TR target genes and the cell cycle remains uncertain. The direct activation of *Ccnd1*, encoding cyclin D1, provides a hypothetical link between T3 and proliferation. This intertwining might also occurs through the regulation of cell cycle gene regulators, as already demonstrated for CREB and AP-1 (*Pascual and Aranda, 2013*).

Thus, inefficient handling of metabolic fuels, reduction of the tissue plasticity, and defect in thermogenic biochemical pathways can explain the apparent default in BAT thermogenesis. This can be compensated by an increased energy consumption which could explain why BATKO mice gained less weight than CTRL mice at room temperature, which represents a mild cold challenge (*Škop et al., 2020*). This extra-energy consumption might favor the onset of other thermogenic mechanisms, as observed with the exacerbated WAT browning. However, WAT browning's ability to consume a significant amount of energy has recently been called into question (*Challa et al., 2020*). Although we failed to measure a striking difference in BATKO mouse oxygen consumption at room temperature over a 48-hr period, a minor increase in the respiratory quotient might have long-term consequences. At

thermoneutrality, where no compensatory mechanisms are required to maintain body temperature, BATKO mice became hypersensitive to diet-induced obesity. This reflected the alteration of diet-induced BAT thermogenesis and the inability of this tissue to optimally use metabolic substrates and thus expend energy. A similar phenotype was previously observed for other mouse models deficient for BAT thermogenesis (*Castillo et al., 2011*).

In humans, BAT has triggered a lot of interest for its ability to increase energy expenditure, which could be used in the fight against obesity. Interestingly, T3 levels also correlate with BAT activity in humans (*Bredella et al., 2012*; *Lahesmaa et al., 2014*). Hyperthyroid patients have higher glucose uptake in BAT, and this effect is neutralized after treatment to restore the euthyroid state (*Lahesmaa et al., 2014*). Therefore, T3 also plays a role in human BAT thermogenesis. Whether the mechanisms regulated by T3 in human BAT are the same as described here in mice is an open question. While T3 cannot be used per se to increase energy expenditure in humans due to cardiac side effects (*Biondi et al., 1993*), its hybridization to other molecules could allow to target it selectively to BAT, as already done for liver (*Finan et al., 2016*).

In conclusion, we used a combination of both omics data and BAT-specific TR-signaling alteration in mice to introduce an unprecedented view of T3 cell-autonomous response in brown adipocytes. We showed that T3 signaling in BAT controls both key metabolic pathways and tissue plasticity that are essential for adaptive thermogenesis. These results represent a valuable database that pave the way for further metabolic studies to detail the molecular implications of T3 signaling during BAT adaptive thermogenesis. Finally, as T3 acts in many other tissues, a similar approach could be used to put together the puzzle of T3 influence in energy expenditure.

## Limitations of the study

One of the objectives of this study was to define a catalog of TR direct target genes in brown adipocytes, based on a combination of RNAseq and ChIPseq. Both TRα and TRβ are present in the BAT (*Minakhina et al., 2020*) but we only performed the TRα1 ChIPseq due to the absence of adequate tools for the TRβ ChIPseq. The cistromes of the two receptors might not fully overlap, and we might have missed a fraction of the TRBS on the genes of interest. However, this limitation should be tempered since previous data have indicated that TRβ-selective binding site are infrequent in the mouse genome (*Chatonnet et al., 2013*). Another limitation is that we assigned genes to a given TRBS by considering only its distance to the transcription starting site. Chromatin 3D organization and insulation are well known to influence the interactions between distant regulatory elements, and the linear distance is only partially informative. The additional genomic investigations required to overcome these limitations remain for the moment hardly feasible in mouse tissues. Finally, we did not look at protein changes while post-translational events could occur.

# Materials and methods

**Key resources table**

| Reagent type (species) or resource | Designation | Source or reference | Identifiers | Additional information |
|---|---|---|---|---|
| Strain, strain background (male mice) | C57BL/6J (male) | Charles River | | |
| Genetic reagent (male mice) | C57BL/6J-BATKO (male) | This paper | | See 'Material and methods', section 'Genetically modified mouse models' |
| Genetic reagent (male mice) | C57BL/6J-BATβKO (male) | This paper | | See 'Material and methods', section 'Genetically modified mouse models' |
| Antibody | Anti-UCP1 (rabbit polyclonal) | Abcam | Abcam: ab10983 | 1:400 |
| Antibody | Anti-rabbit linked to peroxidase (goat polyclonal) | Promega | Promega: W401B | 1:300 |
| Antibody | Anti-tyrosine hydroxylase (rabbit polyclonal) | Merck | Merck: AB152 | 1:500 |
| Antibody | Anti-rabbit IgG linked to peroxidase (goat polyclonal) | Bio-Rad | Bio-Rad: STAR124P | 1:5000 |

*Continued on next page*

*Continued*

| Reagent type (species) or resource | Designation | Source or reference | Identifiers | Additional information |
|---|---|---|---|---|
| Sequence-based reagent | See *Table 2* | This paper | | |
| Commercial assay or kit | Pierce BCA Protein Assay Kit | Thermo Fisher Scientific | Thermo Fisher Scientific: 23225 | |
| Commercial assay or kit | Clarity Western ECL Substrat | Bio-Rad | Bio-Rad: 1705060 | |
| Commercial assay or kit | Diaminobenzidine staining | Sigma-Aldrich | Sigma-Aldrich: D5905 | |
| Commercial assay or kit | Click-iT EdU Cell Proliferation Kit | Thermo Fisher | Thermo Fisher: C10337 | |
| Commercial assay or kit | RNA SENSE Kit | Lexogen | | |
| Commercial assay or kit | Accel-NGS 2S Plus DNA kits | Swift Biosciences | | |
| Chemical compound, drug | Tamoxifen | Sigma-Aldrich | Sigma-Aldrich: T5648 | |
| Chemical compound, drug | PTU-containing diet | Harlan Teklad | Teklad Custome Diet: TD95125 | |
| Chemical compound, drug | T3 | Sigma-Aldrich | Sigma-Aldrich: T2877 | |
| Chemical compound, drug | T4 | Sigma-Aldrich | Sigma-Aldrich: T2376 | |
| Chemical compound, drug | 6-OHDA | Sigma-Aldrich | Sigma-Aldrich: H4381 | |
| Other | Cobas 600 | Roche | | T3/T4 serum quantification |
| Other | Leica TP1020 | Leica | | Tissue section dehydration |

## Animal procedures

All experiments were carried out in accordance with the European Community Council Directive of September 22, 2010 (2010/63/EU) regarding the protection of animals used for experimental and other scientific purposes. The research project was approved by a local animal care and use committee (C2EA015) and authorized by the French Ministry of Research.

## Genetically modified mouse models

The genetic background of all mice that were used in the present study was C57BL6/J. *Thra^{AMI/+}* (***Quignodon et al., 2007***), *Thrb^{lox/lox}* (***Winter et al., 2009***), and *Ucp1^{CreERT2}* (***Rosenwald et al., 2013***) mouse lines were crossbred to introduce the different recombinant alleles in *Ucp1^{CreERT2}xThra^{AMI/+}Thrb^{lox/lox}* mice. The expression of the *Thra^{AMI}* allele allows the expression of the TRα1^{L400R} mutant, which has dominant-negative properties over TRα and TRβ, after Cre/loxP-mediated excision of a stop cassette (***Quignodon et al., 2007***). Despite the persistence of one intact *Thra* allele in *Thra^{AMI/+}* mice after Cre-recombinase action, the dominant-negative action of TRα1^{L400R} eliminates the capacity of cells to respond to T3. *Thrb^{lox}* has 2 tandem-arranged loxP sequences, allowing Cre-mediated excision of exon 3 of the *Thrb* gene, which encodes the DNA-binding domain of the TRβ1/TRβ2 receptor, resulting in a frameshift and a loss of function (***Winter et al., 2009***). In the present study, we used mice of the *Ucp1^{CreERT2}* line, in which CreERT2 expression is under the control of the *Ucp1* promoter (***Rosenwald et al., 2013***). In the absence of physiological stressors, *Ucp1* is specifically expressed in brown adipocytes, thus restricting Cre-mediated recombination to this cell type. CreERT2-recombinase action occurs after its translocation to the nucleus, allowed by tamoxifen treatment. Tamoxifen was injected intraperitoneally every day during 5 days at 50 mg/kg of mice.

In *Ucp1^{CreERT2}* mice, we used a *Rosa26^{TdTomato}* reporter transgene, also known as Ai9 (***Figure 1—figure supplement 1***; ***Madisen et al., 2010***), we verified the absence of recombination activity outside BAT, except for a small fraction of cells present in the choroid plexus. In the following, mice with the *Ucp1^{CreERT2}xThra^{AMI/+}Thrb^{lox/lox}* genotype were called BATKO. *Thra^{AMI/+}Thrb^{lox/lox}* littermates, also

injected with tamoxifen, were used as controls (CTRL). To explore the specific of TRβ for the observed phenotypes in BATKO mice, we also used $Ucp1^{CreERT2}xThrb^{lox/lox}$ called BATβKO mice. $Thrb^{lox/lox}$, also injected with tamoxifen, were used as controls.

In order to address chromatin occupancy by TR specifically in brown adipocytes, we used $Thra^{GS/+}$ mice (*Hirose et al., 2019*) to generate ad hoc $Ucp1^{CreERT2}xThra^{GS/+}$ transgenic mice. These mice were generated by knocking in the *Thra* locus a sequence encoding TRα1 fused with protein G and strepta-vidin protein (GS) after a floxed stop cassette. In the presence of the $Ucp1^{CreERT2}$, tamoxifen injection allows the recombination at the loxP sites that excise the stop cassette specifically in brown adipo-cytes, allowing the expression of the GS-TRα1 only in this cell type. This strategy has already been successfully used in other tissues, including striatum and heart (*Richard et al., 2020*; *Hirose et al., 2019*).

## Experimental animal procedures

We used 2- to 5-month-old male mice for experiments. Genetically modified mice were generated in our own animal facility, whereas wild-type C57BL6/J mice were ordered from a commercial supplier (Charles River). Mice were fed ad libitum with LASQC Rod16 R diet (Altromin, Germany) and housed under recommended conditions (notably, at room temperature, i.e., 23°C). Hypothyroidism in adult animals was induced as previously described, with 14 days of treatment with a propylthiouracil (PTU)-containing diet (Harlan Teklad TD95125, Madison, WI) (*Weiss et al., 1998*). It was combined in some cases by hyperthyroidism induced by intraperitoneal injections of a T3/T4 mix (T4 at 2 µg/g of mice and T3 at 0.2 µg/g of mice, Sigma-Aldrich), daily for the five last days or once at day fourteen of the PTU treatment. T3, the active compound, was not injected alone to get close to hyperthyroid condi-tions met in vivo.

Before assessment of cold response, mice were housed for 10 days at 30°C (with normal 12 hr light/dark cycles) and subcutaneously implanted with IPTT-300 transponders (Plexx BV, Netherlands). The mice were then housed in pairs without enrichment and placed at 4°C during the indicated period. Infrared thermography was performed on awake mice after 48 hr of cold exposure, using an infrared camera (FLiR Systems, Inc). For cell proliferation assessment, EdU (Thermo Fisher) was intraperitone-ally injected twice (100 mg/kg, after 24 and 48 hr of cold exposure) and mice were killed after 72 hr of cold exposure. This allowed to stain proliferative cells over the last 48 hr of cold exposure.

At the end of experiments, mice were anesthetized by an intraperitoneal injection of xylazine (25 mg/kg) and ketamine (130 mg/kg) mixture. Blood was drawn from the vena cava and collected in heparin-coated tubes to retrieve plasma. Several tissues and organs were dissected and either directly processed for histology or snap frozen and stored at −80°C for later RNA preparation.

## Indirect calorimetry and body composition measurement

Body composition was measured in awake mice by low-field nuclear magnetic resonance with a Minispec LF90II device (Bruker). Phenomaster metabolic cages were used for indirect calorimetry measurements (TSE Systems, Berlin, Germany). Mice were first placed in individual cages for a 24-hr period of habituation to isolation, after which oxygen consumption ($VO_2$) and carbon dioxide rejection ($VCO_2$) were continuously recorded for 48 hr, under a normal 12 hr light/dark cycle. $VO_2$ and $VCO_2$ were expressed as ml/hr/kg of mice. Respiratory quotient was obtained as the $VO_2$ / $VCO_2$ ratio. The metabolic rate was calculated according to the Weir formula (*Weir, 1990*) as following: Metabolic rate (kcal/min) = $3.94*VO_2 + 1.1*VCO_2$.

## BAT denervation

Chemical denervation of BAT sympathetic nerve endings was performed in wild-type PTU-fed mice under isoflurane 2% anesthesia and ketoprofen (1 mg/kg) analgesia. This technique, which involves injections of 6-hydroxydopamine (6-OHDA), a neurotoxin selective for sympathetic neurons, permits a specific denervation of sympathetic neurons, while keeping the sensory fibers intact (*Nitta et al., 1992*). The effects of sympathetic denervation were compared to those obtained after vehicle injec-tion (0.15 mol/l NaCl and 1% ascorbic acid, sham mice). Briefly, the two lobes of interscapular BAT were exposed through a midline skin incision along the upper dorsal surface and gently separated from the skin with surgical forceps. Then, injections of 6-OHDA (Sigma-Aldrich) were performed directly into each lobe of the interscapular BAT. For each lobe, 10 µl (10 mg/ml) was injected in several

times (10 injections of 1 µl) using a Hamilton syringe (i.e., 20 µl/mice). The skin incision was then closed with several surgical stitches. Animals were allowed to recover for 5 days before further experiments.

## Western-blot analysis

Cell extracts from BAT were lysed in standard lysis buffer (20 mM Tris–HCl, pH 8, 138 mM NaCl, 1% NP40, 2.7 mM KCl, 1 mM $MgCl_2$, 5% glycerol, 5 mM EDTA (ethylenediaminetetra-acetic acid), 1 mM $Na_3VO_4$, 20 mM NaF, 1 mM dithiothreitol, 1% protease inhibitors), and homogenized using FastPrep (MP Biomedicals). Proteins were assayed in triplicate with Pierce BCA Protein Assay Kit (Thermo Fisher Scientific). Aliquots of 30 µg of proteins, denatured in buffer (20% glycerol, 10% β-mercaptoethanol, 10% sodium dodecyl sulfate [SDS], 62.5 mM Tris) were analyzed from 12% SDS–polyacrylamide gel electrophoresis (PAGE) and transferred to PVDF Immobilon membranes (Bio-Rad). After 1-hr saturation in TBS (Tris-Buffered Saline)/0.2% Tween/2% milk at room temperature, the membranes were probed (overnight at 4°C) with rabbit polyclonal anti-tyrosine hydroxylase (Merck, AB152) diluted in TBS/0.2% Tween/2% milk. Then, membranes were rinsed threetimes in TBS/0.2% Tween for 10 min and incubated for 1 hr with goat secondary anti-rabbit IgG linked to peroxidase (dilution 1:5000; Bio-Rad) in TBS/0.2% Tween/2% milk. Membranes were rinsed again and exposed to Clarity Western ECL Substrate (Bio-Rad). The intensity of the spots was determined by densitometry with ChemiDoc Software (Bio-Rad) and analyzed using the Image LabTM software (Bio-Rad). Quantification of whole protein levels (using the stain-free protocol provided by Bio-Rad) was used for normalization. Stain-Free technology enabled fluorescent visualization of 1D SDS–PAGE gels and corresponding blots. The relative amount of total protein in each lane on the blot was calculated and used for quantitation normalization.

## Histology

BAT and WAT samples were fixed in 10 ml Zinc Formal Fixx (Thermo Fisher Scientific) during 24 hr at 4°C. Samples were dehydrated using the Leica TP1020 semi-enclosed processor and embedded in paraffin. 6 µm sections were processed for immunohistochemistry (IHC) and EdU detection. Tissue sections from different mice were assayed on the same slide to minimize staining variability.

For IHC, deparaffinized sections were incubated overnight at 4°C with rabbit poly-clonal antibodies directed against UCP1 (Abcam, ab10983), diluted 1:400 in phosphate-buffered saline (PBS)/2.5% goat serum. Sections were then incubated with a horseradish peroxidase-labeled anti-rabbit antibody

**Table 2.** Primer sequences.

| Gene | Primer forward | Primer reverse |
|------|----------------|----------------|
| Ucp1 | AAGCTGTGCGATGTCCATGT | AAGCCACAAACCCTTTGAAAA |
| Hr | AGAGGTCCAAGGAGCATCAAGG | TTCCTCTTGTTGCTCTGCCTCC |
| Fndc5 | ATGAAGGAGATGGGGAGGAA | GCGGCAGAAGAGAGCTATAACA |
| Bmp8b | ACCTGTACCGTGCCATGACG | CGGTCGCGTTCCACTATGTTG |
| Idh3a | AAGAGGTTTTGCTGGTGGTGTTC | TTGCGCTCCTCCCACTGAATAG |
| Fabp3 | CATCGAGAAGAACGGGGATA | TCATCTGCTGTCACCTCGTC |
| Dio2 | AACAGCTTCCTCCTAGATGCC | ATTCAGGATTGGAGACGTGC |
| Cidea | GGACAGAAATGGACACCGGGTAG | TGACATTGAGACAGCCGAGGAAG |
| Cox7a1 | AGGCTCTGGTCCGGTCTTTTAG | GGTCATTGTCGGCCTGGAAG |
| Plin5 | GCGCCACACAGCAGAATGTC | GGCAAAGCCACCACTCGATTC |
| Slc25a20 | TCAGGCTTCTTCAGGGGAGAAC | CCACTGGCAGGAACATCTCG |
| Pdk4 | TTTCCAGGCCAACCAATCCAC | GTGGCCCTCATGGCATTCTTG |
| Prdm16 | TAGCTGCTTCTGGGCTCAAGG | ACGTCACCGTCACTTTTGGC |
| Aco2 | TGCCTAAGGTGGCTGTACCATC | CACTTCCTGGTTTATGTCCTTGGC |
| Thrb | CTCTTCTCACGGTTCTCCTC | AACCAGTGCCAGGAATGT |

(1:300, Promega, W401B) for 1 hr at room temperature. Peroxidase activity was visualized with diaminobenzidine staining (Sigma-Aldrich, D5905). Images were acquired using an AxioObserver Zeiss microscope at a ×16 magnification.

EdU detection was assessed as recommended by the manufacturer (Click-iT EdU Cell Proliferation Kit, Thermo Fisher), following deparaffinization of BAT sections. Images were acquired on a DM6000 Leica microscope. Both EdU-positive cells and DAPI (4',6-diamidino-2-phenylindol)-marked nuclei were quantified on pictures of whole BAT sections to avoid any bias of field selection.

## Plasma T3/T4 quantification

Plasmatic free T3 and free T4 were quantified on a Cobas 6000 automat with the Cobas e601module (Roche, ECL analysers).

## RNA extraction and RT-qPCR

RNAs were extracted using Trizol (Invitrogen, Carlsbad, CA, USA). Total RNA was reverse transcribed to cDNA using MMLV reverse transcriptase (Promega, Wisconsin, USA). RT-qPCRs (quantitative reverse transcription polymerase chain reactions) were performed using SYBRGreen mix (Bio-Rad iQ supermix). The results were analyzed according to the ΔΔCT method (*Bookout and Mangelsdorf, 2003*). *Hprt* was used as the reference gene. Primers are listed in *Table 2*.

## RNAseq analysis

cDNA libraries were prepared using the total RNA SENSE kit (Lexogen, Vienna, Austria) and analyzed on a Nextseq 500 sequencer (Illumina) as previously described (*Richard et al., 2020*; *Guyot et al., 2014*). Raw data of single-end sequencing were aligned on the GRCm38 (mm10) reference genome using Bowtie (Galaxy Version 2.2.6.2) and converted to count tables using htseqcount (Galaxy Version 0.6.1galaxy3), respectively. Differential gene expression analysis was performed with DESeq2 (R package, Version 1.34.0) (*Love et al., 2014*) using the following thresholds: adjusted p value <0.05; average expression >10 reads per million, $\log_2$ fold-change >0.6 or <−0.6. The effects of mutations on cold exposure were assessed using the interaction model (*Ge, 2021*). Thresholds for the interaction model were the following: average expression >10 reads per million, adjusted p value <0.05.

Differentially expressed genes expression was visualized as clustered heatmaps using Pheatmap R package (RRID:SCR_016418). Normalized counts from DESeq2 were used as inputs and the correlation method was used to cluster the genes. Data were scaled independently for each gene, with the same color code for all genes (red: above mean, white: mean, blue: below mean).

Gene ontology analyses were made using the Gene Ontology Resource (http://geneontology.org).

## ChIPseq and analysis

Freshly dissected small pieces of BAT were incubated during 25 min at room temperature under agitation with a crosslink solution (250 mM disuccinimidyl glutarate, 50 mM 4-(2-hydroxyethyl)-1-pi perazineethanesulfonic acid, 100 mM NaCl, 1 mM EDTA, 0.5 mM ethylene glycol tetraacetic acid). Then, 1% formaldehyde was added for 20 min followed by 50 mM glycine addition. The tissue was rinsed several times with cold PBS. Chromatin immunoprecipitation was then performed as previously described (*Chatonnet et al., 2013*). Sequencing libraries were prepared from the immunoprecipitated fraction and the input fraction as a control, using the Accel-NGS 2S Plus DNA library kits with single indexing (Swift Biosciences). They were analyzed on a Nextseq 500 sequencer (Illumina). Raw data of paired-end sequencing were aligned on the GRCm38 (mm10) reference genome using Bowtie (Galaxy Version 2.2.6.2). MACS2 (Galaxy Version 2.1.1.20160309.0) was used for peak calling and peaks with a score inferior to 60 were filtered out. De novo motif search was performed using SeqPos motif tool (version 1.0.0). Genes within 30 kb of peaks were called out using GREAT (http://great.stanford.edu/public/html/). We chose a distance of 30 kb upstream or downstream of the TSS to attribute a TRBS to a gene. Although arbitrary, this distance was found to maximize the ratio of T3-responsive genes among the included genes, without excluding genes which have been well characterized as TRα1 target genes in other neural systems, such as *Klf9* or *Hr* (*Gil-Ibañez et al., 2013*). The distribution of distances of TRBSs around TSSs, as well as the distribution of TRBSs in the genome, were assessed using PAVIS (https://manticore.niehs.nih.gov/pavis2/).

## Statistical analysis

The data shown represent the average values for animals with the same genotype that were given the same treatment. The number of animals used in each experiment (*n*) is indicated in figure legends. Except when anything else is mentioned, the error bars represent the standard deviation. For comparing two means, the statistical relevance was determined using an unpaired Student's *t*-test. For determining the effects of the mutations on food consumption, body temperature, body weight overtime, we used a two-way analysis of variance (ANOVA) with time as one factor, and the other parameter as the second factor. For comparing the different levels of one factor to a control group, the statistical relevance was determined using the one-way ANOVA method. When this test showed significant differences (p value <0.05), a post hoc Tukey test was used for multiple comparisons. To assess the effects of the mutations or the denervation on a given treatment (2 factors with 2 levels each), the statistical relevance was determined using a two-way ANOVA. Only when the interaction term was significant (interaction p value <0.05), a post hoc Tukey test was used to compare the effects of the indicated combinations. For any of these tests, statistical relevance is shown in the graphs as follows: *p value <0.05, **p value <0.01, ***p value <0.001.

## Acknowledgements

We thank Catherine Cerrutti and Gerard Benoit for the advices about sequencing analysis as well as Benjamin Gillet, Sandrine Hughes, and the PSI platform of the IGFL for deep sequencing. We also thank Emmanuel Quemener from Center Blaise Pascal/ENSL for the development and maintenance of the ENS Galaxy portal with the help of SIDUS (Single Instance Distributing Universal System). We acknowledge the contribution of the SFR Santé Lyon-Est (UCBL, UAR3453/CNRS, US7/Inserm) facility ANIPHY for their help and the contributions of the CELPHEDIA Infrastructure, especially the center AniRA in Lyon and the Equipex, ANR-11-EQPX-0035 PHENOCAN. We thank Nadine Aguilera and the ANIRA-PBES facility for the mice transgenesis and breeding. We thank internship students Eleonora and Mathilde for their implication at the early steps of the project. We also thank Marine and Manon for their involvement in denervation experiments. Finally, we thank C Wolfrum for providing *Ucp1*<sup>CreERT2</sup> mice. This work was supported by the European Union's Horizon 2020 research and innovation program under grant agreement no. 825753 (ERGO).

## Additional information

### Funding

| Funder | Grant reference number | Author |
|---|---|---|
| European Union's Horizon 2020 | 825753 | Frédéric Flamant |

The funders had no role in study design, data collection, and interpretation, or the decision to submit the work for publication.

### Author contributions

Yanis Zekri, Conceptualization, Resources, Data curation, Software, Formal analysis, Supervision, Validation, Investigation, Visualization, Methodology, Writing – original draft, Writing – review and editing; Romain Guyot, Data curation, Investigation, Methodology; Inés Garteizgogeascoa Suñer, Laurence Canaple, Data curation, Formal analysis, Investigation; Amandine Gautier Stein, Justine Vily Petit, Denise Aubert, Sabine Richard, Data curation, Methodology; Frédéric Flamant, Conceptualization, Resources, Formal analysis, Supervision, Funding acquisition, Validation, Investigation, Visualization, Methodology, Writing – original draft, Project administration, Writing – review and editing; Karine Gauthier, Conceptualization, Resources, Data curation, Formal analysis, Supervision, Validation, Investigation, Visualization, Methodology, Writing – original draft, Project administration, Writing – review and editing

### Author ORCIDs

Yanis Zekri http://orcid.org/0000-0003-4925-4610

Frédéric Flamant [ID] http://orcid.org/0000-0002-3360-2345

### Ethics

All experiments were carried out in accordance with the European Community Council Directive of September 22, 2010 (2010/63/EU) regarding the protection of animals used for experimental and other scientific purposes. The research project was approved by a local animal care and use committee (C2EA015) and authorized by the French Ministry of Research.

### Decision letter and Author response

Decision letter https://doi.org/10.7554/eLife.81996.sa1
Author response https://doi.org/10.7554/eLife.81996.sa2

## Additional files

### Supplementary files

• MDAR checklist

### Data availability

The raw sequencing data and aligned read counts generated as part of this study have been deposited to the NCBI Sequence Read Archive. Accession number: GSE201136; https://www.ncbi.nlm.nih.gov/geo/query/acc.cgi?acc=GSE201136.

The following dataset was generated:

| Author(s) | Year | Dataset title | Dataset URL | Database and Identifier |
|---|---|---|---|---|
| Zekri Y, Guyot R, Flamant F | 2022 | Target genes of thyroid hormone (TH) in brown adipose tissue (BAT) and their role during metabolic stressors | https://www.ncbi.nlm.nih.gov/geo/query/acc.cgi?acc=GSE201136 | NCBI Gene Expression Omnibus, GSE201136 |

The following previously published dataset was used:

| Author(s) | Year | Dataset title | Dataset URL | Database and Identifier |
|---|---|---|---|---|
| Chang JS | 2018 | A map of the PGC-1α- and NT-PGC-1α-regulated transcriptional network in brown adipose tissue | https://www.ncbi.nlm.nih.gov/geo/query/acc.cgi?acc=GSE110053 | NCBI Gene Expression Omnibus, GSE110053 |

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
