## [Editor Report]

This valuable manuscript describing the local action of thyroid hormone on brown adipocytes in adaptive thermogenesis in a mouse model is potentially important in advancing our understanding of thyroid hormone action on adipocytes. The experimental approach used in this paper is solid and the claims are convincingly supported by the data. The strength of the data relies on the genetically modified mice developed by the authors and genetic manipulations employed to achieve selective inactivation of TR in adult BAT to arrive at their findings.

---

## [Decision Letter]

**Decision letter after peer review:**

Thank you for submitting your article "Brown adipocytes local response to thyroid hormone is required for adaptive thermogenesis in adult male mice" for consideration by *eLife*. Your article has been reviewed by 3 peer reviewers, one of whom is a member of our Board of Reviewing Editors, and the evaluation has been overseen by a Reviewing Editor and Mone Zaidi as the Senior Editor. The reviewers have opted to remain anonymous.

Essential revisions

The reviewers have discussed their reviews with one another, and the general agreement is that work is potentially important to the field and well carried out. However, the manuscript cannot be accepted in its current form unless the comments enclosed here are fully addressed. We look forward to receiving a revised version addressing the criticisms. Please follow all recommendations to authors and provide a point-by-point rebuttal.

*Reviewer #2 (Recommendations for the authors):*

This is outstanding work from an established investigator on the molecular mechanisms regulated by T3 in BAT during cold exposure and high-fat diet. The strategy used is novel (selective inactivation of TRs in BAT in adult mice) and uses genetically modified mice developed by the investigators. The analysis presented identified groups of thermogenic genes that were differentially regulated in the BATKO mice (T3 signaling inactivated). These genes are typically involved in thermogenic processes, including multiple mitochondrial pathways. Genes involved in BAT proliferation were also identified. Overall this is a strong manuscript and I have no objections to the methodology used or to the conclusions it reached. What is missing is highlighting the major advances obtained with this study. For example, the fact that TR inactivation was selective to BAT and occurred in the adult BAT is a big deal, which differentiates this study from everything else already published. T3 is important for BAT development and thus the present study bypassed this. I would spell this out to the readers.

The historical context should be revised. The introduction should say that 3 decades ago D2-generated T3 in BAT was shown to transcriptionally accelerate the transcription of UCP1 gene (PMID: 3192531). This was tested in vivo and in isolated brown adipocytes. Extensive work on the UCP1 gene promoter was done a TREs were identified. Thus, the idea that it is not known how T3 stimulates BAT thermogenesis needs revision. Yes, there are other genes as shown here but UCP1 is the key to BAT thermogenesis.

I find it interesting that the two most important genes identified were UCP1 and PGC1a. Both genes are also stimulated by T3 during BAT development. How do the authors interpret this?

*Reviewer #3 (Recommendations for the authors):*

As noted this study provides significant new information on TH signaling in brown adipocytes and the physiologic relevance thereof.

Specific Comments:

1. it is not clear or I missed it. What age was TH signaling knocked out in mice? It mentions mice between 2-5 months old. Were there age-related differences??

2. Would the expected phenotype differ between mice that expressed the dominant negative α allele and the BATKO presented? It is surprising that the authors did not look at specific β deletion alone. No ER CRE alone control with tamoxifen is shown. Was it tested?

Is TRa400 expression similar to WT α expression ie in control versus mutant mice? Finally, does the expression of the GS α occur on the Β KO background or in the presence of β?

3. In Figure 2D it is interesting that the BAT KO mice lose repression of target genes in hypothyroidism. This would imply that the TRalpha allele may lose the ability to bind in the absence of T3. Did the authors look at TRalpha binding in hypothyroidism?

4. In Figure 3 no attempt was made to define what stimulated food intake. Given the potential paracrine or endocrine control via BAT it might be important to discern if any orexigenic pathways were altered to mediate this phenotype. Also, the mechanisms around Figures 3C and D needs to be clarified. This looks similar to some of the phenotypes seen in D2KO mice but it is not well clarified here. Was food intake measured?

5. In Figure 4, presumably the partial response seen in the BAT KO mice is due to the SNS. A denervation experiment here would have been important.

6. Figure 5 looks at BAT proliferation but "browning" is not explored or the mechanism therein.

---

## [Author Response]

Reviewer #2 (Recommendations for the authors):This is outstanding work from an established investigator on the molecular mechanisms regulated by T3 in BAT during cold exposure and high-fat diet. The strategy used is novel (selective inactivation of TRs in BAT in adult mice) and uses genetically modified mice developed by the investigators. The analysis presented identified groups of thermogenic genes that were differentially regulated in the BATKO mice (T3 signaling inactivated). These genes are typically involved in thermogenic processes, including multiple mitochondrial pathways. Genes involved in BAT proliferation were also identified. Overall this is a strong manuscript and I have no objections to the methodology used or to the conclusions it reached. What is missing is highlighting the major advances obtained with this study. For example, the fact that TR inactivation was selective to BAT and occurred in the adult BAT is a big deal, which differentiates this study from everything else already published. T3 is important for BAT development and thus the present study bypassed this. I would spell this out to the readers.

We thank the reviewer for these kind remarks and excellent suggestion. It is true that many important aspects and progresses that differentiate our study from others were not sufficiently highlighted. We added a section about this at the beginning of the ‘discussion’ section, and we think that it considerably reinforces the outcome of our work (see lines 247-250).

The historical context should be revised. The introduction should say that 3 decades ago D2-generated T3 in BAT was shown to transcriptionally accelerate the transcription of UCP1 gene (PMID: 3192531). This was tested in vivo and in isolated brown adipocytes. Extensive work on the UCP1 gene promoter was done a TREs were identified. Thus, the idea that it is not known how T3 stimulates BAT thermogenesis needs revision. Yes, there are other genes as shown here but UCP1 is the key to BAT thermogenesis.

We are sorry if we omitted to consider the historical context appropriately and the suggested articles are now referenced. Yet, we want to be careful: UCP1 is indeed key to BAT thermogenesis. However, these observations are mostly based on UCP1-KO animals that present altered mitochondrial function (Kazak et al., 2017, PNAS). In the absence of functional mitochondria, any UCP1-independent mechanism involving mitochondria would not be altered. The phenotype of UCP1-KO mice is thus not directly related to UCP1 metabolic function, but rather reflect a developmental alteration of BAT. Other authors have highlighted UCP1-independent mechanisms, like creatine futile cycle, which inhibition decreases brown adipocytes thermogenesis and energy expenditure even in presence of UCP1 (Rahbani et al., 2021, Nature). The current trend is to consider that UCP1-independent mechanisms also bring a significant contribution to thermogenesis.

Based on the reviewer recommendation, we modified the introduction to remind the known link between T3, UCP1 and BAT thermogenesis (lines 76-78) but that a full description of the T3 function in BAT based on transcriptomic data and appropriate animal models was missing (see line 86-88).

I find it interesting that the two most important genes identified were UCP1 and PGC1a. Both genes are also stimulated by T3 during BAT development. How do the authors interpret this?

This is an interesting point. UCP1 and PGC1a are up-regulated early in life in BAT as a result of local T4 deiodination (Hal et all, 2010, Endocrinology). T3 might be link to the post-natal BAT expansion (Negron et al., Sci Rep 2020) and thus also promote adipocytes proliferation in both juveniles and adults. At adult stages, the chromatin environment may be similar and T3 is most likely to regulate an overlapping set of genes. This is something that we recently observed in a compilation of thyroid hormone target genes in several mice tissues: a huge majority of genes upregulated by T3 in the striatum at P15 is also upregulated at adult stages (Zekri et al., 2022, IJMS).

Reviewer #3 (Recommendations for the authors):As noted this study provides significant new information on TH signaling in brown adipocytes and the physiologic relevance thereof.Specific Comments:1. It is not clear or I missed it. What age was TH signaling knocked out in mice? It mentions mice between 2-5 months old. Were there age-related differences??

Indeed, TH signaling was knocked-out in 2-5 months old mice. As we combined 3 different transgenes with different levels of zygosity, we had small cohorts and it was tricky to obtain mice within the same age at the same time. Importantly, this heterogeneity occurred both in CTRL and BATKO groups, allowing to neutralize age-related interferences. We completely agree that this is an important issue. Based on your remark, we compared the transcriptome of mice according to their age. In this new analysis, we split animals in two groups according to their age: the youngest and the oldest (independently of the treatment). DESeq2 analysis evidenced that 9 genes were differentially expressed between these two groups, but none of them were genes that we sorted out as interesting in our publication. This evidences that our results are not contaminated by the ‘age’ factor. But you were completely right, this is something that we should have checked before and this is now something we will systematically check on RNAseq data. Thank you very much for this constructive remark.

2. Would the expected phenotype differ between mice that expressed the dominant negative α allele and the BATKO presented? It is surprising that the authors did not look at specific β deletion alone. No ER CRE alone control with tamoxifen is shown. Was it tested?

You are right, due to our model combining knock-in for TRα and knock-out for TRβ, we cannot distinguish which effects are TRα or TRβ-dependent. Despite not being the focus of our study, this is a really interesting question. We have not questioned the effects of TRα^AMI^ alone as it presents dominant-negative effects towards TRβ. However, we have partially established the phenotype of mice with a selective deletion in brown adipocytes of TRβ only. These mice present a similar phenotype as BATKO mice, since they gain less weight than CTRL mice at room temperature when fed a high fat diet. They are also cold-sensitive in the absence of food. Thus, at least a part of BATKO phenotype relies on TRβ. However, we must not forget that TRα is also expressed and has been shown to control specific function in BAT thermogenesis (Marrif, Endocrinology, 2005). As we wanted to determine the whole T3 response and did not want to deal with the residual response of TRα in TRβKO mice, we focused here our attention on BATKO mice. BATKO mice give a more representative picture of T3 signaling in brown adipocytes.

We agree with you that this is an important aspect. Thus, we introduced in this publication mice with the deletion of TRβ only (called BATβKO). We mentioned their phenotype in the main text and we added the corresponding figures as supplementary data (see lines 174-177, lines 182-185, Figure 3 —figure supplement 2 and the associated legend lines 897-901).

Characterization of the Ucp1-Cre has been done by the lab that generated this Cre (Rosenwald, Nature Cell Biology, 2013). This paper was cited in the method section, but we also added it in the Results section when presenting the model (line 99).

Is TRa400 expression similar to WT α expression ie in control versus mutant mice?

Yes, this is something that can be seen on the Sanger sequencing made on cDNA from BATKO mice (Figure 1B). Even if this method is only semi-quantitative, we don’t observe any tendency for TRa400 to more expressed than the WT allele. However, it is difficult to recognize if the wild-type allele is similarly expressed between CTRL and BATKO mice.

Finally, does the expression of the GS α occur on the Β KO background or in the presence of β?

Indeed, GS-TRα binding occurs even in the presence of ΤRβ as GS-TRα1 is expressed in wild-type mice. As GS-TRα1 is expressed under the control of the wild-type TRα1 promoter, it is a good indication that TRα1 is significantly expressed in BAT.

Previous in vitro work of our team (Chatonnet, 2013, PNAS) highlighted that TRα1 and TRβ1 most likely bind on similar sites. In the context of this study, we acknowledge that introducing GS-TRα in TRβKO would be a convincing indication that it extends, or not, the binding sites of GS-TRα. This idea could perfectly fit a future study that aims at deciphering the respective roles and binding specificity of TRα1 and TRβ1.

3. In Figure 2D it is interesting that the BAT KO mice lose repression of target genes in hypothyroidism. This would imply that the TRalpha allele may lose the ability to bind in the absence of T3. Did the authors look at TRalpha binding in hypothyroidism?

Indeed, we looked at TRα binding in hypothyroidism conditions. These data have been gathered with others in a recently published atlas of thyroid hormone target genes (Zekri et al., 2022, IJMS). We observed that T3 treatment slightly decreases the number of TRα binding sites in brown adipocytes, but the majority of TRα binding sites are shared between hypo- and hyperthyroid conditions. However, it is difficult to draw conclusions from these comparisons as they are based on only one sample. This is also not a general model as T3 treatment increases, rather than decreases, the number of TRβ binding sites in the liver (Shabtai, 2021, Genes Dev).

Concerning TRα^AMI^: one explanation would be that the small subset of genes you highlighted in figure 2H corresponds to TRβ-regulated genes, where the mutated TRα does not play a dominant-negative effect. Without the repression neither mediated by TRα^AMI^ nor TRβ in the absence of T3, the expression of these genes “leaks” in the mutants, explaining the differences observed in figure 2H. The hypothesis that TRα^AMI^ cannot bind DNA in the absence of T3 is not the most likely, as the number of genes which expression ‘leaks’ would be much more important. One hardly-feasible way to test that would be to combine TRα^AMI^ allele with the GS-tag, and compare its binding sites with GS-TRα1.

4. In Figure 3 no attempt was made to define what stimulated food intake. Given the potential paracrine or endocrine control via BAT it might be important to discern if any orexigenic pathways were altered to mediate this phenotype. Also, the mechanisms around Figures 3C and D needs to be clarified. This looks similar to some of the phenotypes seen in D2KO mice but it is not well clarified here. Was food intake measured?

In response to cold exposure, we hypothesized that food intake is secondary to the higher energy expenditure required for BATKO animals to maintain their body temperature (Figure 3A), as already described in hypothyroid animals. In the absence of food, the increased energy expenditure cannot occur and BATKO animals’ temperature drops significantly (Figure 3B). Unfortunately, the device we used for indirect calorimetry cannot be used at 4°C to experimentally test this hypothesis. We agree that this would have been a very interesting demonstration.

During HFD feeding, the food intake was measured and was not different between BATKO and CTRL mice, a phenotype indeed similar as the one observed after the inhibition of D2 in adipose tissues (Fonseca, 2014, Diabetes). Thus, food intake is not the reason why BATKO mice gain more weight than CTRL mice. You are right to mention it because it is a very important information to interpret this phenotype. Thus, we added figures to show these data. They are presented in figure 3C and 3D, and mentioned at lines 181 and 190 with the associated legend at lines 853-860.

Thank you very much for that, it will clearly help the reader to better understand the results.

5. In Figure 4, presumably the partial response seen in the BAT KO mice is due to the SNS. A denervation experiment here would have been important.

You are completely right, this is the most probable hypothesis. For example, Ucp1 expression becomes less sensitive to norepinephrine in hypothyroid rats. On the contrary, BAT denervation prevents the optimal increased expression of Ucp1 mediated by T3 (Bianco, 1988). Thus, the SNS and T3 act synergically to regulate Ucp1 expression. The absence or alteration of one system still allow a residual response to the other one. More broadly, DIO2-KO mice mount an exaggerated response of the SNS to compensate inefficient BAT thermogenesis (Christoffolete, 2004, Diabetes). Thus, the residual response is most likely a consequence of an important SNS stimulation of the BAT. The intertwining between T3 signaling and the SNS is a very complex question that would deserve a complete study in the future.

6. Figure 5 looks at BAT proliferation but "browning" is not explored or the mechanism therein.

Indeed, figure 5 focuses on BAT proliferation during a 72h cold exposure. This experiment was made to reinforce the transcriptomic observations and we did not aim at understanding the underlying mechanisms. Recent publications were specifically dedicated to this task (Liu, 2022, Nat Comm) and are referenced in the main text (line 299) if it triggers the readers curiosity.

Concerning WAT browning, we have presented the data in Figure 3 —figure supplement 3. WAT browning was observed after 3 days of cold exposure in CTRL mice, but this phenomenon was exacerbated in BATKO mice. Please note that this event is unrelated to the T3 response in the WAT as TR are not mutated in this tissue in BATKO mice.

A massive WAT browning was also observed in UCP1-KO mice (Ukropec, 2006) or in mice with surgically suppressed BAT (Piao, 2018). As the above-cited authors, we interpreted this as a compensatory mechanism to generate heat in the absence of an efficient BAT thermogenesis. However, we must be careful about this hypothesis as the contribution of WAT browning to thermogenesis and energy expenditure as recently been called into question (Johann, 2019 / Challa, 2020).